# Modelling biology in novel ways - an AI-first course in Structural Bioinformatics

**Kieran Didi**
University of Cambridge/Heidelberg
ked48@cam.ac.uk

**Charles Harris**
University of Cambridge
cch57@cam.ac.uk

**Pietro Lio**
University of Cambridge
pl219@cam.ac.uk

**Rainer Beck**
University of Heidelberg
rainer.beck@bzh.uni-heidelberg.de

## Abstract

In recent years, there has been tremendous progress in applying data-driven methodologies to study biological questions. The rapidly evolving field of machine learning has gained a plethora of methods that can be applied to structural biology like protein structure prediction. However, the intricacies one faces when analyzing complex biological data are sometimes underappreciated in applications of machine learning methods. On the other hand, biologists often face a language- and method barrier when trying to understand and correctly apply machine learning tools. As a result, they might be using such methods without proper expertise, potentially resulting in incorrect predictions and questionable conclusions about the resulting data. To help remedy these issues, we have developed a holistic 11-unit course in AI-driven Structural Bioinformatics with the aim of (i) encouraging machine learning researchers to learn more about the biological complexity of the data they are analyzing and (ii) allowing biologists to better understand state-of-the-art machine learning algorithms for correct application to biological systems. The course includes video lectures, animated visualisations as well as in-depth exercises and further resources for each of the topics discussed. We hope that our course stimulates collaboration across research communities and lowers the entry barrier for newcomers to understand and investigate structural biology with data-driven tools. Our course is available at https://structural-bioinformatics.netlify.app.

## 1 Introduction

The application of machine learning (ML) to structural biology has causesd significant advancements in biological problem-solving, particularly in protein structure prediction [1] and *de novo* protein design [2, 3]. However, applying ML tools to biology is complicated, often neglecting the specific challenges and subtleties inherent to modelling biological data and leading to issues in real-world application scenarios [4, 5, 6, 7]. The barrier to effective understanding and application of machine learning in biology is intensified by a distinct disconnect between the two fields. On one hand, biologists grapple with the technical language and methodologies inherent to ML tools. On the other, ML researchers frequently lack the vital biological context that should inform their research. This divergence in expertise can culminate in well-intended but misdirected solutions.

Although bioinformatics courses are available [8], a gap remains in integrating the swiftly evolving domain of ML applied to bioinformatics, specifically in the feild of structural bioinformatics. To bridge this gap, we have developed an 11-unit AI-driven Structural Bioinformatics course. This course

NeurIPS 2023 AI for Science Workshop.

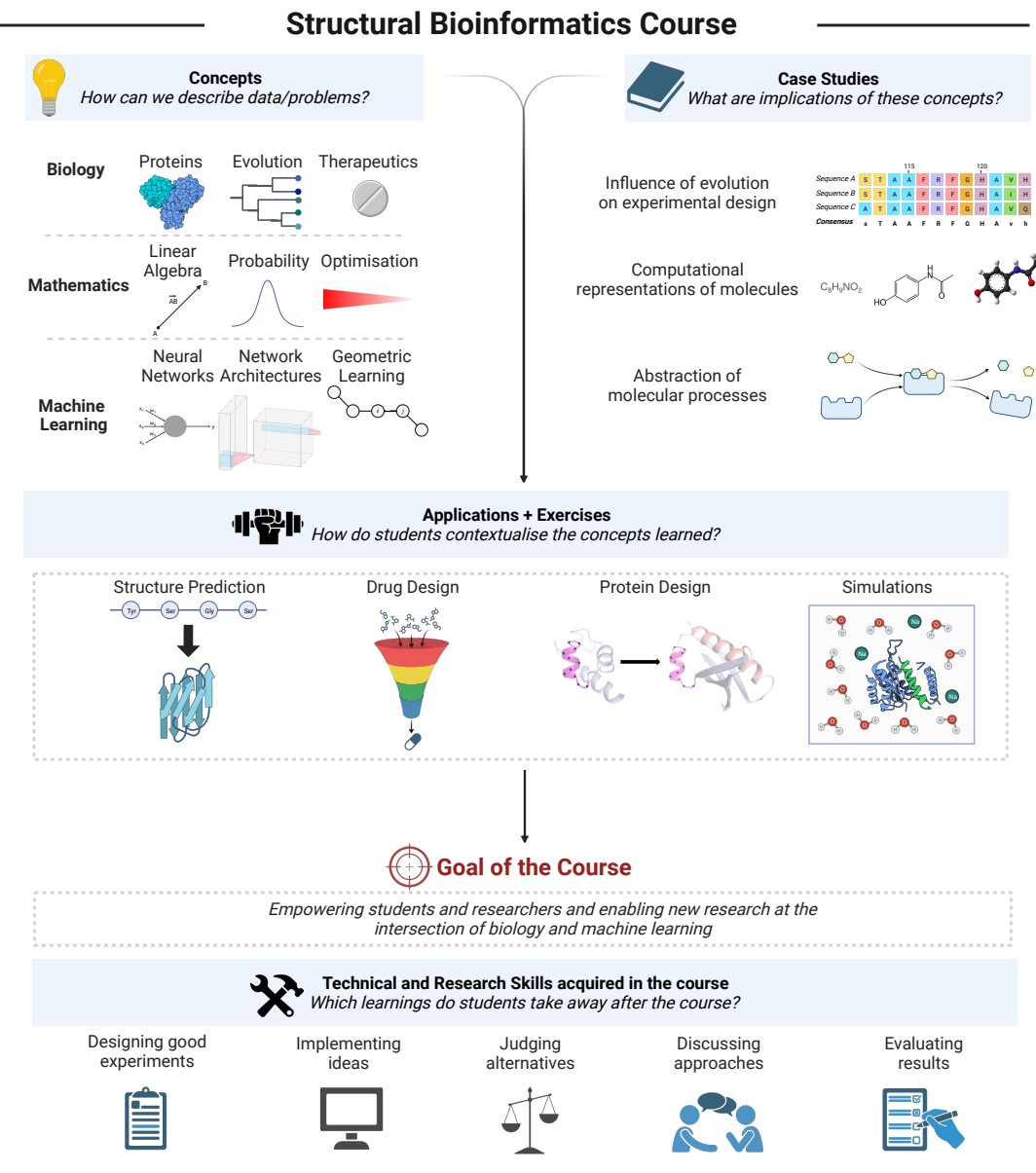

Figure 1: Course overview. Students get exposed to conceptual ideas in three different disciplines and learn about their interactions and implications via case studies. They use the taught concepts in exercises on practical applications and by this gain technical as well as research skills to investigate the intersection of biology and machine learning.

aims to equip ML researchers with biological understanding and enable biologists to understand and apply the latest ML research effectively, fostering interdisciplinary understanding and collaboration.

## 2  Background

The exact definition and curriculum of bioinformatics have been subjects of long-standing discussions [9, 10], which have gained renewed significance due to the advancements AI has brought to the field, enabling exploration of biological datasets in novel ways.

Bioinformatics, a multidisciplinary field, intertwines biology, mathematics, and computer science to interpret biological data [11]. Given its rapid evolution [12], creating up-to-date curricula is essential to equip students with both theoretical knowledge and practical skills, addressing the significant skill gap in life sciences [13, 14]. The integration of programming competencies like Python or Bash scripting is fundamental for applying theoretical knowledge in practical settings [15].

**Curriculum Design and Interdisciplinary Approach**   Curricula need to facilitate knowledge consolidation through projects and case studies, enabling contextual application of acquired knowledge [16]. Considering the interconnectedness of courses within broader student curricula [17], this course adopts a model in which the core curriculum explores the interface of biology and machine learning in depth and primers are provided to accommodate diverse backgrounds and enables learners to fill in any existing knowledge gaps. Different courses set their focus differently (App. B); our course aims to integrate knowledge from machine learning and biology instead of teaching the two in isolation.

An interdisciplinary approach, involving educators from life sciences and computer science, is pivotal for a cohesive learning environment [18], preparing students for the future interdisciplinary demands and the integration of AI technologies in advancing structural bioinformatics.

## 3  Course Contents

The course aims to empower both students and researchers to dive into AI applications in structural biology and is structured into three main parts to achieve this goal (Fig. 1). Areas within structural bioinformatics such as protein structure prediction, evolutionary modelling, molecular dynamics simulations, early stage drug design and *de novo* protein design are introduced. In each area, both the traditional methods (e.g. docking software) and the latest machine learning methods (e.g. deep learning-based docking) are introduced, allowing for the comparison of similarities and differences as well as exploration of intricacies with are handled poorly by current ML methods [4, 6]

To provide the appropriate background knowledge to understand current applications, concepts from three areas are taught: biology, mathematics and machine learning. To offer students a curriculum specific to their needs, many of these fundamental concepts (e.g. linear algebra or probability) are presented as primers, meaning they are taught in a self-contained lesson at a suitable point in the curriculum and can be shortened for students who already possess the required knowledge. A detailed overview of the course content and concepts can be found in the Appendix in Fig. 3.

The concepts from these three domains often seem disconnected at first. That is why we integrate case studies into the curriculum; in our course, the students can see the connections between seemingly disparate concepts and how one concept can be applied to solve problems related to a different one. Examples include the choice of computational representation of molecules that is heavily influenced by the task at hand [19, 20] and also the influence of evolution on experimental design and methodologies such as the difficult of crafting robust train-test splits of biological data [4, 21]

Mixing passive with active learning elements promises to engage students more in the learning process and improve learning outcomes [22]. Therefore, we augment our concept lectures and case studies with exercises. In these hands-on problems, students learn to apply the learned concepts to an interesting problem of practical relevance such as drug design or molecular simulation and directly see the usefulness of concepts learned in the lectures.

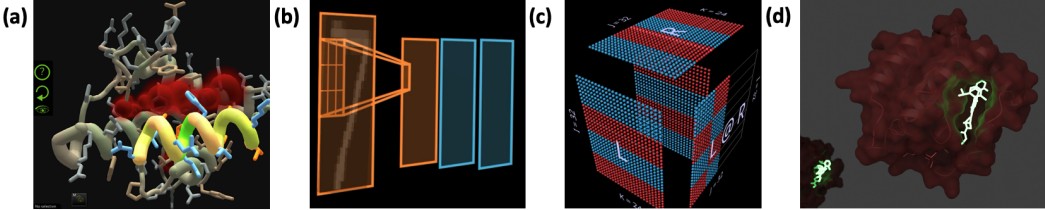

Figure 2: The curriculum includes both interactive exercises like (a) the FoldIt protein design game as well as interactive animations for concepts like (b) convolutions or (c) matrix multiplication. Students will also learn how to present their results with tools like (d) PyMol and Molecular Nodes [29].

## 4 Teaching Concept and Techniques

**Curating and Developing Teaching Materials**   In alignment with guidelines for curriculum development in bioinformatics, the course strategically leverages existing teaching materials, curating and adapting them to serve the comprehensive educational objectives [23]. This didactically appropriate approach ensures the representation of varied contents and the development of new materials for underrepresented topics. Such a combination of curated and freshly developed resources, which are made publicly available, supports other educators in enhancing their instructional materials [23].

Specifically, the course incorporates adapted notebooks and packages from the TeachOpenCADD platform [24] and visualization tutorials [25], ensuring a rich and diversified learning experience.

**Role of Conceptual Diagrams and Visualizations**   Conceptual diagrams are crucial to teaching complex concepts, but their creation poses considerable challenges and time investments [26]. The course addresses this by integrating a blend of distinguished visualizations available and custom-made diagrams and examples, aimed at providing clarity and enhancing comprehension (Fig. 2).

To visually illustrate the protein design process, the course employs PyMol [27], allowing students to interactively explore the intricacies of protein structures. This interaction is further enriched by additional tutorials based on the FoldIt tool [28], which gamifies the learning experience, fostering intuitive understanding and engagement.

**Visualization of Machine Learning Concepts**   The course incorporates ManimML [30] to animate machine learning concepts such as convolutions or variational autoencoders, offering students intuitive insights into these advanced topics. This tool, in combination with the mm tool [31], facilitates the visualization of fundamental concepts like matrix multiplications, enabling students to grasp concepts related to efficient training and model interpretability effectively.

Additionally, Penrose [32] is deployed to generate clear diagrams for mathematical concepts like conditional independence and Bayes rule, ensuring that students gain a clear understanding of the mathematical underpinnings essential for bioinformatics.

## 5 Conclusion

In this paper, we detail an 11-unit course focused on AI-driven Structural Bioinformatics, constructed with the aim of addressing the notable divide between biology and machine learning disciplines. The curriculum is carefully developed, considering the complex nature of biological data and the intricacies of modern machine learning methods.

In summary, this course endeavours to provide a balanced perspective on the intersection of biology and machine learning, promoting interdisciplinary understanding and collaboration. The availability of course materials serves as a resource for those seeking to explore the possibilities and challenges in the evolving landscape of structural bioinformatics.

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

## A The curriculum in detail

In Fig. 3, we break the individual lessons of the course down into the tree main subject disciplines and which parts of the course teach what concepts. In addition, the case studies we look at are described. The actual course content can be found at the course website `https://structural-bioinformatics.netlify.app`.

**Structural Bioinformatics, detailed curriculum**

| Lecture | Biology | Mathematics | CS/Machine Learning | Case Studies |
|---|---|---|---|---|
| L1: Introduction | Protein structure, history of the field | Intro to linear algebra + probability | Biological file formats + handling | PDB files |
| L2: ML Basics | - | Optimisation, gradient descent | Neural networks, basic notions | PyTorch |
| L3: ML Architectures | Computational representation of proteins | Matrix Algebra | CNNs, RNNs, transformers | AlexNet, transformers |
| L4: Language, Evolution and Bioinformatics | Homology, phylogeny | Distance metrics, clustering | Language models, data leakage | ESM |
| L5: Geometric Deep Learning | Computational representation of generic molecules | Invariance, equivariance, group theory | Graph Neural Networks (GNNs), geometric graph learning | GCN, GAT, EGNN |
| L6: Protein Structure Prediction | Structure-Function relationship, coevolution, protein dynamics/interactions | End-to-end differentiability, quaternions | Inductive biases in model building, self-supervised learning | AlphaFold2, ESMFold |
| L7: Generative Modelling | - | distribution learning, score functions | Function modelling vs generative modelling, VAEs, diffusion models | Autoregressive VAEs, DDPMs |
| L8: Protein Design | Sequence- vs structure-based methods, catalysis, functional motifs | SO(3) group equivariance | Equivariant diffusion models | Rosetta, RFDiffusion, ProteinMPNN |
| L9: Simulations | Protein dynamics, conformational flexibility, structure ensembles | Numerical vs analytical integration, Newton's equations of motion | Performance/accuracy trade-off, coarse-graining, multiprocessing | GROMACS, Allegro |
| L10: Drug Design | Protein-ligand interactions, virtual screening | - | Rephrasing a problem as a generative one, data-driven vs rule-based methods | AutoDock, DiffDock, DiffSBDD |
| L11: Further Topics and Conclusion | Summary and Conclusion | Summary and Conclusion | Summary and Conclusion | - |

Figure 3: The course curriculum in more detail. The lesson contents and learning outcomes are separated into the three main subject categories (biology, mathematics, CS/Machine Learning); in addition, the case studies discussed that illustrate the different concepts during the lesson are mentioned.

## B Related courses

The course by Vater et al. integrates a design-to-data workflow in a biochemistry Course-based Undergraduate Research Experience (CURE), connecting students to a global community of protein researchers, thus enriching the undergraduate research landscape [33]. Yang et al. explored a cohort-based learning approach to improve computational protein design literacy among undergraduates, providing valuable research opportunities during the COVID-19 pandemic [34]. Le et al. developed a hands-on education strategy with sixteen modules using PyRosetta Jupyter notebooks to teach biomolecular structure and design, covering topics from conformational sampling to protein docking, and RNA structure prediction [35].

Centeno et al. described a course that employs hands-on computer approaches to teach structural biology, preparing students to build protein models based on sequence and structure information [36]. Engelberger et al. crafted cloud-based tutorials for distance learning on structural bioinformatics, combining bioinformatics software, interactive coding, and visualization exercises, which proved beneficial during the remote learning necessitated by the COVID-19 pandemic [37]. Menke et al. introduced a course on artificial intelligence and deep learning using interactive electronic programming notebooks, aiming to provide a hands-on learning experience in AI and deep learning fields [38]. Clune et al developed a 1-week mathematics boot camp for incoming chemistry graduate students and provided exercises and lecture materials for foundational mathematical topics like probability or differential equations [39].

