# OpenReview forum: "Modelling biology in novel ways -  an AI-first course in Structural Bioinformatics"
_NeurIPS.cc/2023/Workshop/AI4Science — NeurIPS2023-AI4Science Poster_

### Official Review · Reviewer_AM1X · 2023-10-19
**Review of Modelling biology in novel ways - an AI-first course in Structural Bioinformatics**

**Rating:** 7
**Confidence:** 2

**Review:**

This reviewer is not sure of the bar for acceptance in the Education track. Nevertheless, the course outline seems quite reasonable as a collection of topics and would have caught my interest at an earlier career stage. The breakdown of each module in Bio, Math and ML is nice but seems a bit forced at times. The provided website is lacking in materials, but if this is indeed an actual university course to be eventually taught then I think it is worthy of being highlighted at the workshop.

---

### Official Review · Reviewer_dCyk · 2023-10-21
**Fascinating course proposal with some questions about scoping**

**Rating:** 9
**Confidence:** 3

**Review:**

This work describes a course developed to teach students about AI in structural biology. The authors focus on developing a course that is applicable to students with computational and biological backgrounds, and it emphasizes hands-on work with modern AI models while equipping students with tools to understand future developments in this quickly-evolving domain of AI research.

I believe this proposed course would offer a valuable experience for students, and the authors' presentation of course structure and content is incredibly effective. The course has a focus on methodologies to close the gap between computational and biological backgrounds, and it focuses on practical activities to support students' capabilities to implement and utilize these systems. In addition, I am impressed by the focus on interaction with the data in this work; the authors emphasize the importance of developing understanding of the fundamental nature of protein and molecular structures. My only question is around defining a proper scope for the course. The authors propose an ambitious plan to give students the proper scientific and computational background to understand protein structures and ML fundamentals. I worry that there is not enough time devoted to ML fundamentals, which may leave behind students with more biology-oriented backgrounds when studying incredibly technical topics in geometric deep learning. However, I believe a lot of these issues could be resolved by requiring some basic mathematical background as a prerequisite to this course, such as a probability, statistics, or linear algebra course. Alternatively, the technical information presented in latter units could be reduced, with more of a focus on general applicability of the methods.

Overall, I recommend this work for strong acceptance. The authors present what seems to be an effective course to introduce students to AI for structural biology, and the presentation of materials in the paper is incredibly well-written.

---

### Meta-Review · Area_Chair_HCau · 2023-10-26

**Recommendation:** Accept (Poster)
**Confidence:** 4

**Metareview:**

This paper describes a course to teach biologists about AI. The website is easy to use and well organized. The reviewers agree that this course will be of broad interest to the broader community. Recommendation: Poster.